# Genotoxins: The Mechanistic Links between *Escherichia coli* and Colorectal Cancer

**DOI:** 10.3390/cancers15041152

**Published:** 2023-02-10

**Authors:** Ya Wang, Kai Fu

**Affiliations:** 1Institute of Molecular Precision Medicine & Hunan Key Laboratory of Molecular Precision Medicine, Department of General Surgery, Xiangya Hospital, Central South University, Changsha 410008, China; 2Center for Medical Genetics & Hunan Key Laboratory of Medical Genetics, School of Life Sciences, Central South University, Changsha 410078, China; 3Hunan Key Laboratory of Animal Models for Human Diseases, Central South University, Changsha 410008, China; 4National Clinical Research Center for Geriatric Disorders, Changsha 410008, China; 5Hunan Key Laboratory of Aging Biology, Xiangya Hospital, Central South University, Changsha 410008, China

**Keywords:** bacteria, genotoxin, colorectal cancer, *Escherichia coli*, DNA damage

## Abstract

**Simple Summary:**

Colorectal cancer (CRC) is a leading cause of cancer-related mortality worldwide, and its incidence is steadily rising in developing areas. Bacterial pathogens from the gut microbiota have been recognized as important risk factors for CRC. This review updates the current knowledge of *Escherichia coli*-produced genotoxins related to CRC.

**Abstract:**

Emerging evidence indicates bacterial infections contribute to the formation of cancers. Bacterial genotoxins are effectors that cause DNA damage by introducing single- and double-strand DNA breaks in the host cells. The first bacterial genotoxin cytolethal distending toxin (CDT) was a protein identified in 1987 in a pathogenic strain in *Escherichia coli* (*E. coli*) isolated from a young patient. The peptide-polyketide genotoxin colibactin is produced by the phylogenetic group B_2_ of *E. coli*. Recently, a protein produced by attaching/effacing (A/E) pathogens, including enteropathogenic and enterohemorrhagic *E. coli* (EPEC and EHEC) and their murine equivalent *Citrobacter rodentium* (CR), has been reported as a novel protein genotoxin, being injected via the type III secretion system (T3SS) into host cells and harboring direct DNA digestion activity with a catalytic histidine-aspartic acid dyad. These *E. coli*-produced genotoxins impair host DNA, which results in senescence or apoptosis of the target cells if the damage is beyond repair. Conversely, host cells can survive and proliferate if the genotoxin-induced DNA damage is not severe enough to kill them. The surviving cells may accumulate genomic instability and acquire malignant traits. This review presents the cellular responses of infection with the genotoxins-producing *E. coli* and discusses the current knowledge of the tumorigenic potential of these toxins.

## 1. Introduction

Colorectal cancer (CRC) is a leading cause of cancer-related death globally, and its incidence is steadily rising in developing areas [1]. CRCs occur sporadically in the majority of cases, and only 2~5% of cases are related to a hereditary polyposis syndrome [2]. Mutations in the tumor suppressor gene adenomatous polyposis coli (*APC*) located on chromosome 5q21 are considered one of the earliest events in the initiation and progression of CRC, occurring in 70~80% of sporadic CRC. *APC* encodes a multifunctional protein that is involved in multiple processes, including cell cycle control, cell adhesion and migration, proliferation, apoptosis, and chromosome segregation [3], having recently been reported to have key functions in DNA repair regulation [4]. Intestinal tumors can spontaneously form in *APC*-related mouse models, which have been designed to contain germline mutations in the *APC* gene that result in expression of a truncated APC protein. Conversely, familial adenomatous polyposis (FAP) caused by heterozygous *APC* germline mutations accounts for only 1% of CRC. Lynch syndrome (LC), accounting for 2~4% of cases, is the most common cause of inherited CRC. LC is caused by the genetic defect in one of the genes, most commonly *MSH2*, *MLH1*, *MSH6*, and *PMS2*, that are responsible for mismatch repair (MMR) [5]. MMR enhances DNA replication fidelity and/or reduces mutations caused by DNA damage [6]. Deficiency of MMR enhances somatic mutation of tumor suppressors such as *APC* and *TP53*, which are mutated genes in the majority of CRCs [7]. The genetic evidence suggests that excessive DNA damage may be an important cause of CRC if the damaged DNA was not fixed properly.

Microbial pathogens, mostly viruses and bacteria, cause a wide variety of human diseases, including cancer. Microbial infection has been considered to contribute to 20% of all types of tumors in humans [8]. Besides genetic mutations, several parameters including diet, inflammatory processes, and bacterial pathogens from the gut microbiota have been recognized as important risk factors for CRC [9,10]. Although emerging evidence suggests the direct bacterial contributions to cancer formation, it has not been extensively investigated yet as to how bacteria participate in tumorigenesis. Various bacterial species have been associated with colonic tumorigenesis, such as *Streptococcus bovis*, *Bacteroides fragilis*, *Enterococcus faecalis*, *Clostridium septicum*, *Fusobacterium* spp., and *Escherichia coli* (*E. coli*), part of the normal intestinal flora [8]. *E. coli*, a diverse group of Gram-negative, facultatively anaerobic, rod-shaped bacteria of the family Enterobacteriaceae, is a component of the normal flora in the human gut. However, some strains are pathogenic. Intestinal pathogenic *E. coli* can be traditionally divided into six pathotypes on the basis of their pathogenicity profiles: enteropathogenic *E. coli* (EPEC), enterohemorrhagic *E. coli* (EHEC), Shiga-toxin-producing *E. coli* (STEC), enteroaggregative *E. coli* (EAEC), enterotoxigenic *E. coli* (ETEC), and diffusely adherent *E. coli* (DAEC) [11]. A new pathotype named adherent invasive *E. coli* (AIEC) has been reported by Boudeau et al. [12]. *E. coli* strains are classified into main phylogenetic groups (A, B_1_, B_2_, and D) on the basis of phylogenetic analysis [13]. Phylogenetic groups A, B_1_, and D are responsible for intestinal infections, whereas groups B_2_ and D are responsible for extraintestinal infections [14,15]. Pathogenic *E. coli* strains own pathogenicity islands that encode virulence genes that promote bacterial invasion and colonization, as well as evasion from the host immune responses, thus resulting in damage to eukaryotic host cells and systemic dissemination, thereby allowing for the establishment of infection in hosts. Pathogenic *E. coli* possess many different virulence factors important for colonizing, invading the host, and evading host immune defenses [16,17], some of which, especially genotoxins, have been associated with initiation or progression of CRC. Genotoxins are agents or chemicals that can cause DNA or chromosomal damage, thereby causing mutations. This DNA damage in a somatic cell could result in somatic mutations, which may lead to the malignant transformation of the cell and cancer eventually [18]. The first genotoxin from pathogenic *E. coli* named cytolethal distending toxin (CDT) was reported to be involved in human CRC [19,20]. Another genotoxin, colibactin, encoded by the polyketide synthases (*pks*) pathogenicity island of *E. coli*, has been linked to CRC, and colibactin-related mutational signatures were found in CRC patients [21,22]. Virulence factors delivered by the type III secretion system (T3SS) are typically called effectors. Of note, UshA, one of the effectors injected into host cells through T3SS by some pathogenic *E. coli* and their murine equivalent *Citrobacter rodentium* (CR), was identified as a novel genotoxin recently, which digests different forms of DNA and promotes the colonic tumorigenesis [23].

Thus far, only three DNA-damaging toxins produced by *E. coli* have been identified. This review will focus on these genotoxins, which is of particular interest because of their key roles in initiation and promotion of CRC, thus detection and removal, as well as reducing the chance of exposure to certain strains of *E. coli*, could decrease the risk of cancer in a large group of individuals.

## 2. Cytolethal Distending Toxin

### 2.1. Structural Features and Subunits of Cytolethal Distending Toxin

The CDT was firstly identified by Johnson and Lior in 1987 as a novel heat-labile toxin from pathogenic *E. coli* strains [24]. Several decades past, CDT has been reported in more than 30 other Gram-negative pathogenic bacteria, being responsible for major foodborne diseases all over the world [25]. Unlike other bacterial protein toxins such as hemolysin A and Shiga toxin, which act either on the plasma membrane or the target present in the cytosol, CDT functions by causing damage to the cell’s DNA.

All members of the CDT family are AB_2_ heterotrimers consisting of the active subunit CdtB (A) and two binding moieties, CdtA and CdtC (B), of which CdtB is the most conserved subunit [25,26]. These three subunits form an active heterotrimeric complex in a ratio of 1:1:1. As shown in Figure 1 [27], CdtA and CdtC, whose structure shares homology with lectins, form a deep groove and a large aromatic patch, which helps CDT bind to receptor molecules on the lipid membrane of the host cell. The structure of CdtB contains a four-layered fold: stranded β-sandwich flanked with α-helix and loops, which is the characteristic of the DNase I family. The sequence homology of *cdtA*, *cdtB*, and *cdtC* genes from different bacteria species is diverse. Even CDTs produced by the same species may be different, for example, five types (CDT-I to CDT-V) have been identified in *E. coli*. CDT exerts cytotoxic effects as an AB_2_ structure with two regulatory subunits, CdtA and CdtC, mostly responsible for host cell recognition of CDT, as they bind to a receptor on the plasma membrane of host cells [28], responsible for transporting the active subunit CdtB into the nucleus of host cells, which bears the catalytic activity [29,30]. However, the identity of the CDT receptor is still not fully characterized. Several studies have come to conflicting results about the nature of the CDT receptor [31], which includes cholesterol-rich lipid rafts, the sphingomyelin synthase 1 protein, and glycoproteic or glycolipidic cellular receptors that are diverse among the CDTs produced by different species [32]. After binding to certain receptors, CdtB and CdtC are internalized through the endocytic pathway, while CdtA remains on the plasma membrane of the host cells [33]. Alternatively, CDTs are delivered into the host cells via fusion between outer membrane vesicles (OMVs) and the plasma membrane [34]. OMVs, containing periplasmic proteins including CDT in *E. coli* and other bacterial species, result from bacterial membrane budding. OMVs have been demonstrated to be internalized through fusing with lipid rafts of the plasma membrane of the host cells [35]. It reported that CdtB could translocated into the nucleus by hijacking the retrograde transport pathway responsible for the Golgi apparatus and endoplasmic reticulum (ER) [36]. Moreover, brefeldin A, which disrupts retrograde protein transport from the Golgi complex to the ER, has been demonstrated to block CDT-mediated cell cycle arrest [37] (Figure 2). CdtB reaches the host nucleus mainly by an endoplasmic-reticulum-associated degradation (ERAD) pathway and then is translocated into the nucleus through the nuclear membrane [36]. CdtB is translocated from the ER mainly through an ERAD-like pathway, whereas the CdtC protein is degraded in the ER [38]. It is also possible that CdtB, which possesses a nuclear-localization signal (NLS), first enters the cytosol from the ER and then crosses the nuclear membrane via the nuclear pore complex [36].

### 2.2. Carcinogenic Effects of Cytolethal Distending Toxin

The analysis of crystal structures of CDT indicates that CdtB has sequence homology with DNase I family proteins [29,39]. In vitro results show that CdtB is able to cut or relax supercoiled DNA [30,39]. Of note, the CDT effects showed similarities with that of damage evoked by ionizing radiations and activated similar pathways [40]. CDT caused DNA strand breaks activating DNA damage response (DDR), facilitating the activation of the DNA repair pathway mediated by ATM-CHK2 and ATR-CHK1, and in turn led to cell cycle arrest and apoptosis in human cells [41,42]. A detailed analysis of the kinetics and type of DNA damage indicate that low doses of *E. coli*-produced CDT (50 pg/mL) first induce single-strand breaks (SSBs), resulting in a replicative stress and inhibiting the progression of the replicative fork, and eventually converted into double-strand breaks (DSBs) during the S phase [43]. Conversely, high doses of *E. coli*-produced CDT (above 75 ng/mL) majorly result in DBSs, possibly due to the occurrence of juxtaposing SSBs on opposite strands.

The presence of CDT-producing bacteria has been linked to carcinogenesis and to the localization of tumor tissue. For example, the prevalence of mucosal-associated CDT-producing *E. coli* was found in proximity of tumors but not in the healthy part of the colon [19]. Exposure to sublethal doses of *E. coli*-derived-CDT triggers a reduced DDR in *APC*- and *TP53*-deficient colonic epithelial cells [44]. Increased mutation frequency, accumulation of chromatin, and chromosomal aberrations were observed in cells cultured with sublethal doses of CDT for more than 30 weeks [45]. The chronic exposure to the genotoxin CDT was related to impairment of DDR and failure to efficiently activate cell cycle checkpoints, which then resulted in the acquisition of characteristics of malignant tumor in fibroblasts and colonic epithelial cells in vitro [45]. CDT produced by diverse bacterial species lead to different tumorigeneses. As shows in Table 1, CDT-producing *Campylobacter jejuni* (*C. jejuni*) promotes CRC tumorigenesis through inducing DNA damage in *Apc*^Min/+^ mice [46]. Similarly, CDT-producing *Helicobacter hepaticus* (*H. hepaticus*) promotes intestinal carcinogenesis through inducing increased DSBs and activating the Stat3 signaling pathway [47]. Moreover, CDT-producing *H. hepaticus* promotes hepatocarcinogenesis by the activation of the NF-κB pathway in A/JCr mice [48]. These findings suggest that the genotoxin CDT may be a critical virulence protein implicated in tumorigenesis.

## 3. Colibactin

### 3.1. Structural Features of Colibactin

Extraintestinal pathogenic *E. coli* strains usually express a variety of virulence-enhancing factors that confer the capacity to colonize, invade, survive in the blood compartment, and evade the host immune monitoring and cause damage to the host [49]. These virulence genes encoding virulence-enhancing factors can be clustered on pathogenicity islands, genomic regions that have been acquired by horizontal gene transfer. Note that not all virulence genes of *E. coli* are located on pathogenicity island, some are plasmid encoded or simply reside on the chromosome [50].

The 54-kb *pks* genomic island is present in 30 to 40% of phylogenetic group B2 *E. coli* strains [51]. The *pks* genomic island encodes nonribosomal peptide synthetase (NRPS) and polyketide synthetase (PKS), assembling a hybrid PK-NRP compound called colibactin, which is considered to be a bacterial natural product related to human health [52]. Recently, the structure of colibactin has shown that each biosynthetic gene in the colibactin gene cluster is involved in the production of precolibactin, the precursor of colibactin. A precolibactin, the inactive form, is cleaved by an unusual inner membrane-bound periplasmic peptidase ClbP (colibactin peptidase) to trigger the formation of active genotoxins [53,54] (Figure 3). The enzymatic domain of ClbP is located outside the cytoplasm, which allows for the genotoxin colibactin outside the cytoplasm after maturation, probably to protect the genome of colibactin-producing bacteria [54]. Active colibactin is composed of two complex biosynthetic intermediates with a nearly symmetrical structure that contains two electrophilic cyclopropane warheads [55]. Colibactin alkylates DNA with a ‘double warhead’ consisting of a cyclopropane ring combined with an α,β-unsaturated imine in vivo, creating DNA adducts in mammalian cells and germ-free mice [55,56].

### 3.2. Colibactin-Induced Genotoxicity and CRC

Certain colibactin-producing *E. coli* strains are frequently associated with human CRC (55~66.7% vs. 19~21% associated with normal intestinal tissue) [19,57]. These *E. coli* strains induce DSBs, chromosomal rearrangements, mutations, and cell cycle arrest, and have been shown to be carcinogenic in mouse models [52,57,58]. Studies have shown that *pks^+^ E. coli* is prevalent in colon tissue of CRC patients, suggesting the carcinogenic effect of colibactin [59]. Inoculating mice with *pks^+^ E. coli* strain NC101 resulted in DNA damage in colonic epithelial cells (CECs), then promoted tumor multiplicity and invasion [57]. By contrast, genotoxicity induced by the infection of *pks*-deleted *E.coli* mutants, as well as tumor multiplicity and invasion, were reduced [57]. Colibactin has been well documented to cause DSBs in host cells [60]. It has been reported that *pks^+^ E. coli* infection induced not only interstrand cross-links but also ATR-dependent replication stress response in cultured human cells [60]. In addition, it has been reported that different multiplicity of infections (MOIs) of *pks^+^ E. coli* led to diverse outcomes of CRC progression. Xenografts infected by *pks^+^ E. coli* at low MOI (20) promoted tumor growth, while infection at high MOI (100) resulted in reduced tumor growth, compared with uninfected controls [61], the mechanism of which may be that infection at low MOI causes the emergence of senescent cells, which consequently produced growth factors that might promote tumor growth [61]. Studies have shown that colibactin may impact colorectal carcinogenesis [61,62]. Contradictorily, *E. coli* Nissle 1917 containing *pks* island that allows for the production of colibactin exerted a probiotic effect without any reported adverse effects [63]. A non-genotoxic *E. coli* Nissle 1917 mutant also lost its probiotic activity in models of experimental colitis, suggesting that the genotoxicity of *E. coli* Nissle 1917 could not be isolated from its probiotic activity [63]. In summary, our knowledge of colibactin is very limited, and more around the nature of colibactin is yet to be elucidated.

### 3.3. Unique Mutational Signatures Induced by Colibactin

In a recent study, human intestinal organoids were subjected to colibactin-producing *E. coli* strain or *clbQ*-deficient *E. coli* strain that was unable to produce colibactin by repeated luminal microinjections for 5 months. Whole-genome sequencing (WGS) of the organoids showed that colibactin exposure led to two unique mutational signatures, including a unique single-base substitution (SBS88) signature characterized by T > A and T > C mutations, particularly at ATA, ATT, and TTT motifs, as well as an ID (ID18) featuring single A or T deletions at poly(dA:dT) tracts termed ID-*pks* [21]. Encouragingly, results showed that SBS-*pks* and ID-L were significantly enriched in CRC metastases compared with other cancer types, on the basis of an unbiased analysis of WGS data from 3668 human solid cancer metastases (496 CRC cases). In fact, the positive correlation between SBS-*pks* and ID-*pks* suggests that the two mutational signatures have a common origin. Interestingly, *APC*, the most frequently mutated gene in CRC, had the largest number of mutations that matched the motif caused by colibactin (SBS-*pks* and ID-*pks)* [21]. Therefore, colibactin may promote CRC progression in a manner similar to APC mutation (Figure 3). Together, these results elucidate the *pks^+^ E. coli*–induced mutational signature, besides building a strong mechanistic link between colibactin-producing *E. coli* exposure and CRC. A subsequent study has revealed that organoids derived from primary murine colon epithelial cells after short-term exposure to colibactin-producing *E. coli* showed characteristic features of CRC cells, such as Wnt-independence, impaired differentiation, and enhanced proliferation, as well as genomic instability and mutations, suggesting short-term exposure to colibactin is sufficient to transform normal primary epithelial cells into a precancerous state [64]. Short-term exposure to colibactin-producing *E. coli* did not lead to mutations in the classic Wnt-signaling, but in genes associated with P53 signaling pathway, including *Trp53* and *miR-34a* (Figure 3). These findings may help us better understand the carcinogenic mechanism of colibactin.

Some critical issues in the field of colibactin that remain are the prime focus of current research. For example, the genotoxicity of colibactin relies on the direct contact with host cells [52], albeit not only is it unclear as to how the protein enters the cell and moves into the nucleus to find the DNA, but also it has not been explained how the highly active protein avoids firing its warhead on membranes or histones and reserves it for DNA, which is not present as naked DNA in a cell but is tightly packed. It is also unclear but important to explore the benefits that the colibactin provides to the bacteria, which may be related to bacterial colonization, persistence, and survival in host tissues.

## 4. UshA

### 4.1. Biological Functions of UshA

UshA, a bifunctional enzyme that possesses 5′-nucleotidase and UDP-sugar hydrolase activity, has sequence homology to the mammalian ecto-5′-nucleotidases and is highly conserved among bacterial species [65]. The enzymatic functions of UshA are most thoroughly studied in *E. coli*. UshA has been reported to be involved in nucleotide salvage and to be necessary for bacterial growth on 5′-AMP as the only carbon source [66,67]. UshA was also approved to be a major periplasmic enzyme for nicotinamide adenine dinucleotide (NAD) degradation in *E. coli* [68]. Specifically, using its NAD(H) pyrophosphatase activity, UshA hydrolyzed NAD to NMN and AMP, which were further hydrolyzed to nicotinamide riboside (NmR) and adenosine (Ado), respectively, through its 5′-nucleotidase activity. Deletion of the *ushA* gene resulted in faster growth of *E. coli* cells in M9 medium and better stability of exogenous NAD [68]. The N-terminal metallophos domain (Pfam ID PF00149) of UshA contains the catalytic site, including a dimetal center and a catalytic histidine, as well as a C-terminal 5_nucleotid_C domain (Pfam ID PF02872) that contains a substrate-binding site, or specificity site, with two aromatic residues [69,70]. A≈20-amino acid linker joined the N- and the C-domains of UshA together. UshA interacts with the nucleotide substrates by the adenine ring between two phenylalanine residues of the C-terminal domain, and then the C-terminal domain of UshA undergoes a large hinge-bending rotation and it brings substrates to the N-terminal catalytic domain where the reaction takes place [71].

### 4.2. Attaching and Effacing (A/E) Pathogens and Their Tumorigenic Role in CRC

Certain pathotypes of *E. coli* are a major cause of foodborne diseases, which lead to severe economic burden, serious morbidity, and mortality worldwide [72]. EPEC was the first pathotype of *E. coli* associated with human disease and is considered one of the predominant causative agents of human diarrhea in infants in developing countries [11,73]. EHEC is an emerging human pathogen that can cause acute gastroenteritis and hemolytic uremic syndrome in children under 5 years and the elderly [74,75]. Both EPEC and EHEC, as well as the mouse pathogen *Citrobacter rodentium* (CR), elicit transient and noninvasive infections in the host via forming attaching and effacing (A/E) lesions to the intestinal epithelium [76]. CR is the murine equivalent of EPEC and EHEC and is widely used as a model to understand the molecular basis of human-A/E-pathogen-induced acute infection in vivo and in vitro [77,78]. An A/E lesion is characterized by the localized effacement of microvilli and remarkable changes of the cytoskeleton, including the accumulation of polymerized F-actin [79,80,81]. The A/E lesions occur in three stages: (i) initial adherence, (ii) signal transduction, and (iii) intimate attachment [82]. After adherence to the host cells, multiple virulence proteins, also called effectors, are injected into the host cells through T3SS, a highly conserved specialized protein-secretion apparatus by A/E pathogens at the second stage; thereafter, they interfere with a series of signal transduction pathways of cells, which promote pathogen survival and evading immune responses. The number of translocated effectors varies from approximately 22 in EPEC and 39 in certain EHEC strains to as many as 29 in CR [83,84,85,86].

Several research studies have shown a strong link between adherent *E. coli* strains and CRC [87,88]. EPEC can completely deplete host cell DNA mismatch repair (MMR) proteins through secreting the targeted mitochondrial effector protein named EspF and induces the reactive oxygen species (ROS) in a EspF independent manner, thereby reducing DNA repair activity and increasing mutation rates in host cells [89,90]. The *APC*^Min/+^ mouse, which harbors a germline mutation in one allele of *APC* and developed adenomas in the cecum and distal colon spontaneously, is a common model for the early steps of human CRC [91]. Interestingly, preclinical studies demonstrated that CR infection promoted epithelial cell proliferation and increased the colonic adenomas number in *APC*^Min/+^ mice compared with uninfected *APC*^Min/+^ mice. Although tumorigenic potential of A/E pathogens was suggested by early evidence [89,90,92], the underlying mechanisms of colon tumorigenesis, which is promoted by noninvasive infection of the pathogen, remain mysterious. In our recent study, UshA was validated as a virulence protein, which is injected into the host cells via T3SS, then cleaves host DNA and leads to colonic tumorigenesis [23] (Figure 4). However, as a newly discovered genotoxin, more details about the carcinogenesis of UshA need to be further studied. UshA can only enter the host cytoplasm through the T3SS of bacteria, and the way in which UshA enters the nucleus and then cuts the host DNA is unknown. Considering that the molecular weight of UshA is about 63kDa, it cannot enter the host nucleus by passive diffusion, but can only through active transport. Thus, there are two possibilities: (i) UshA may contain a nuclear localization signal (NLS), by which it can be transported into the nucleus. (ii) UshA may interact with a protein containing NLS and then is transported together into the nucleus. Once in the nucleus, how does UshA play its genotoxic role? The mutational signatures of the *APC*^Min/+^ mice model was identified by whole exome sequencing, and the results show that UshA mainly cause single base substitutions (SBS), of which UshA-conferred SBS26 is associated with a defect in DNA mismatch repair and ultimately accelerates colon tumorigenesis (Figure 4) [23]. Whether the genotoxic effects of UshA contribute to tumor initiation by other means remains to be investigated.

### 4.3. UshA as a Newly Identified T3SS-Dependent Genotoxin Related to CRC

CRC has been well documented to be caused by multiple genomic alterations that lead to genomic instability [93]. Our results demonstrate that A/E pathogens possess the intrinsic genotoxic capability [23]. *ΔescN* CR, an isogenic mutant with failure to inject virulence proteins into host cells due to a defective T3SS [94], failed to cause DNA damage compared to wild-type CR in an infected germ-free (GF) mouse model. The data indicate that CR per se directly provokes genotoxic stress to colon during infection in vivo in the absence of normal gut microbiota, consistent with the critical role of T3SS in CR-infection-induced colonic crypt hyperplasia [95]. To identify the T3SS-dependent genotoxin, the SILAC (stable isotope labeling with amino acids in cell culture)-based mass spectrometry data were evaluated on the basis of the putative functions, abundance, and conservation among A/E pathogens, and then 11 proteins were considered as candidate genotoxins. The DNA digestion assays demonstrate that UshA harbors intrinsic DNA damaging capability to single-stranded DNA, linear double-stranded lambda DNA, circular plasmid DNA, and extracted chromatin DNA in vitro. A catalytic histidine–aspartic acid (His-Asp) dyad was proposed to be the main player in the catalytic mechanism of UshA [96]. In contrast to wild-type UshA, CR-UshA^DM^, a mutant with catalytic histidine–aspartic acid dyad substituted with alanine–alanine, almost lost the ability to digest DNA. EPEC-, EHEC-, and non-pathogenic *E. coli* K-12-UshA are well conserved among these strains, displaying comparable ability to digest these substrates. Although expressing functional UshA, *E. coli* K-12 failed to damage host DNA due to lacking T3SS [23].

Of note, a publication showed that CR infection promotes colonic tumor formation in genetically susceptible mice, which may be due to the infection-induced hyperproliferative state [92]. In line with this study, inoculation with CR and *ΔushA* CR complementing with UshA (*ΔushA*::UshA CR), but not *ΔushA* CR and *ΔushA* CR complementing with catalytic-dead UshA mutant (*ΔushA*::UshA^DM^ CR), dramatically accelerated the development of colon adenoma in a genetically susceptible *APC*^MinΔ716/+^ mouse model. Conversely, the colonization, proliferation, and clearance of CR, *ΔushA* CR, *ΔushA*::UshA CR, and *ΔushA*::UshA^DM^ CR were largely comparable after oral inoculation. Consistently, in an ex vivo three-dimensional organoid culture system, the colonic organoids derived from CR- and *ΔushA*::UshA CR-infected colon tissues grew dramatically faster and larger than those from vehicle control- and *ΔushA* CR-infected ones, suggesting that UshA is the essential factor for CR-infection-mediated augment of the stemness and tumorigenic potential of colonic cells [23]. On the basis of whole-exome sequencing (WES), the single base substitution (SBS) analysis of tumors isolated from vehicle control, CR, and *ΔushA* CR-infected *APC*^MinΔ716/+^ mice revealed UshA-dependent unique signatures called SBS26, characterized by high proportions of T > C mutations and was observed in human CRC. Interestingly, SBS26 is a mutational signature associated with defective DNA mismatch repair associated with CRC [97]. The characteristics of the mutational signature are consistent with a previous publication that the A/E pathogen EPEC infection promoted host mutation via depletion of DNA mismatch repair proteins including MSH2 and MLH1. These results highlight that acute and transient infection indued by certain pathogens could harbor a far-reaching impact on the progression of CRC, which is completely different from that of chronic and persistent infection by bacteria, such as Helicobacter pylori and hepatitis B, which promote the development of cancer.

As a novel identified genotoxin, multiple interesting questions need to be answered in the future. For instance, UshA-dependent infection increases the occurrence of SBS26, which is a human CRC mutational signature associated with defective DNA mismatch repair. The roles of UshA in stage or prognosis of CRC still need to be revealed. It is also important to know the beneficial effects that UshA provides to A/E pathogens during the infection. The mechanism by which UshA was transported into the nucleus from the cytosol of host cells remains unknown. In addition, the mechanism by which the genome of UshA-producing bacteria is protected from the genotoxicity of UshA needs to be explored.

## 5. Conclusions and Perspective

Over the last decades, CRC as a common and lethal disease has been a hot topic in both basic and clinical research. While our understanding of the connections between intestinal microbiome and CRC continues to strengthen, the contributions of *E. coli* in this community to initiation and promotion of CRC remain largely unknown. In particular, the roles of genotoxin, which causes DNA damage with consequent genomic instability, produced by *E. coli* in CRC are poorly understood. This review updates the current knowledge of three *E. coli*-producing genotoxins that contribute to CRC, including protein structure, function, and carcinogenic mechanism, as shown in Table 2. In terms of protein structure, CDT is a heterotrimer consisting of three subunits, while colibactin is s polyketide-nonribosomal peptide, and UshA is a bifunctional enzyme with homology to the mammalian ecto-5′-nucleotidases. All three genotoxins can cause DNA damage, but each has its own characteristics. CDT possesses nicking or relaxation activity, as well as colibactin alkylate chromatin DNA, while UshA directly digests DNA substrates in vitro. The entry of genotoxin into the host cell is vital for its carcinogenic function. CDT binds to the receptor on the surface of host cell or is wrapped in outer membrane vesicles, which then fuse with the plasma membrane to transport the toxin into the cell. UshA is transported into the host cell through T3SS. However, the way in which colibactin enters the host cell remains unknown. Studies on the relationship between genotoxins and CRC may provide new ideas for prevention and treatment of this severe disease; however, many questions remain to be answered in the near future.

## Figures and Tables

**Figure 1 cancers-15-01152-f001:**
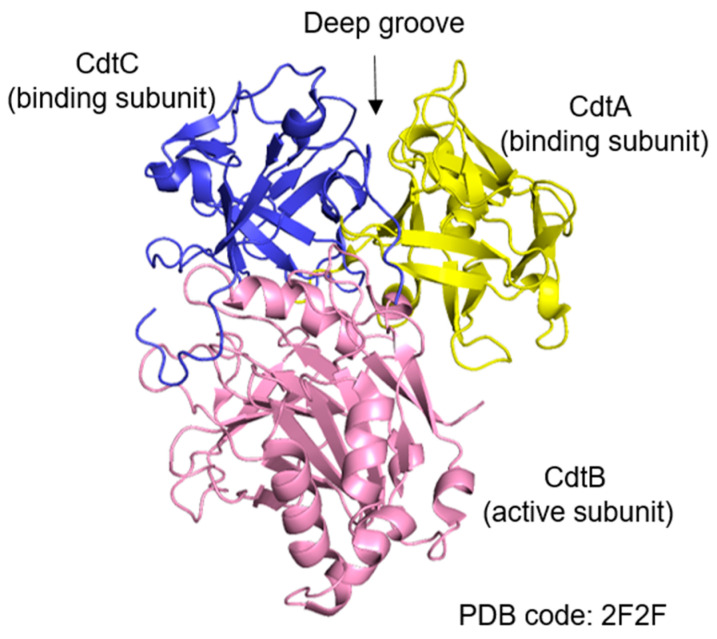
The PDB structure of the CDT toxin from *Actinobacillus actinomycetemcomitans*. The yellow color indicates the CdtA subunit, the pink color is for the CdtB subunit, and the blue color codes for the CdtC subunit. CdtA and CdtC together form a deep grove.

**Figure 2 cancers-15-01152-f002:**
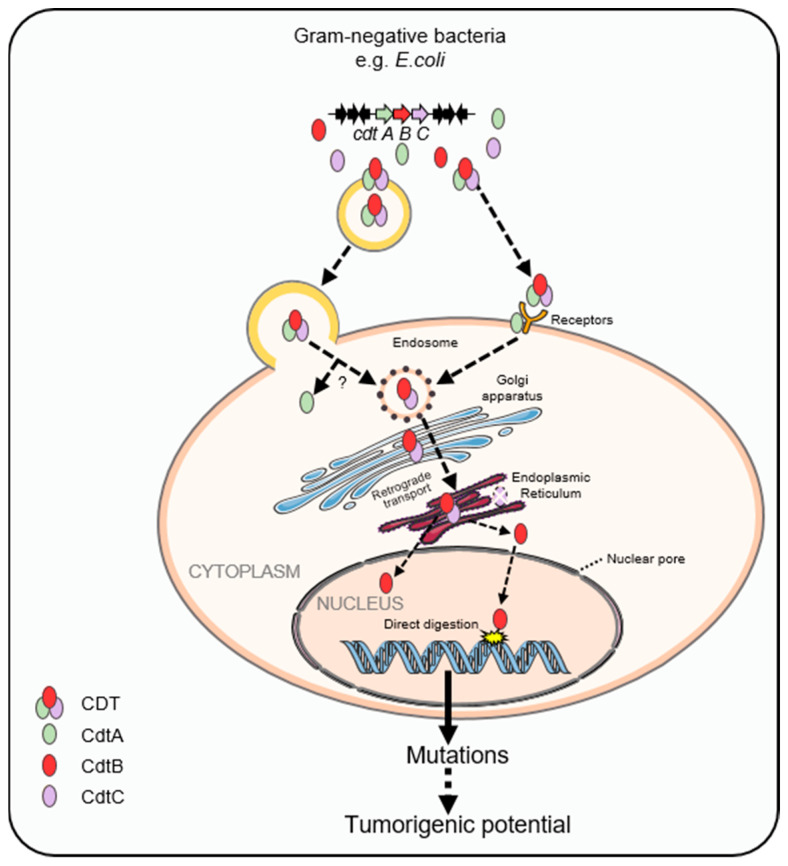
Cytolethal distending toxin enters the host cell and digests DNA in its nucleus. In Gram-negative bacteria, e.g., *E. coli*, the three subunits CdtA, CdtB and CdtC, encoded by *cdtA*, *cdtB*, and *cdtC* genes, respectively, aggregate on chromosomes to form a constitutively expressed operon, which is assembled in the periplasm. After being secreted, CDT binds to the receptor on the surface of the host cell or is wrapped in outer membrane vesicles, which then fuse with the plasma membrane to transport the toxin into the cell. The CdtB–CdtC heterodimers enter the cell through endocytosis. Then, CdtB is retrotransported through the Golgi apparatus via an ERAD-like pathway. CdtB enters the nucleus through a nuclear pore complex, then digests host DNA by using its DNase I activity. The host cells accumulate mutations if the DNA damage is incompletely repaired. Mutations are accumulated in the host cell if damaged DNA is repaired incompletely, which may result in enhanced tumorigenic potential.

**Figure 3 cancers-15-01152-f003:**
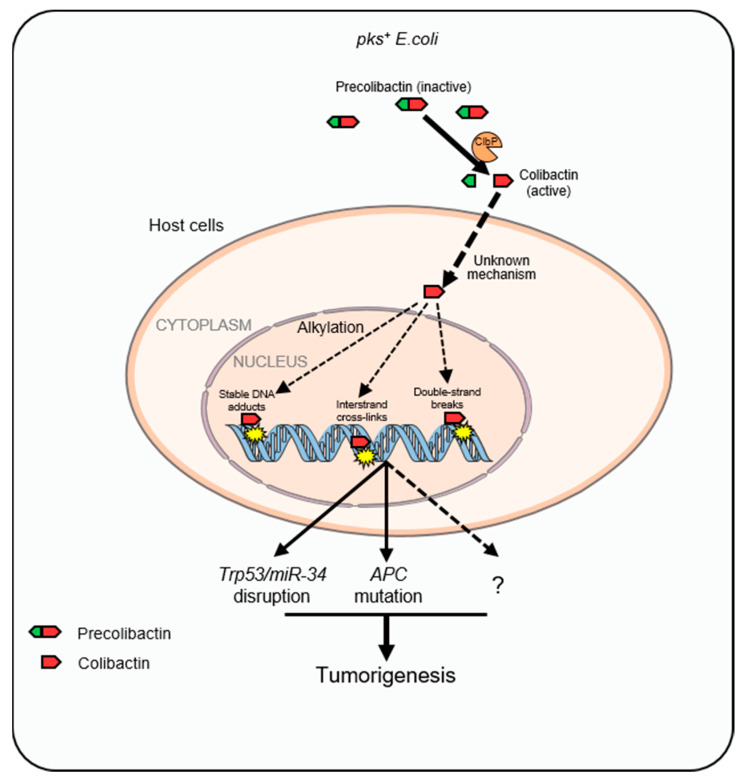
The carcinogenic mechanisms of *pks^+^ E. coli*-produced colibactin. Mature colibactin, produced by *pks^+^ E. coli* after peptidase ClbP-mediated cleavage, is translocated into the nucleus of host cells, then alkylates chromatin DNA and causes specific DNA lesions including double-stranded DNA breaks, stable DNA adducts, and interstrand cross-links. Colibactin-induced DNA insults have been associated with CRC susceptibility.

**Figure 4 cancers-15-01152-f004:**
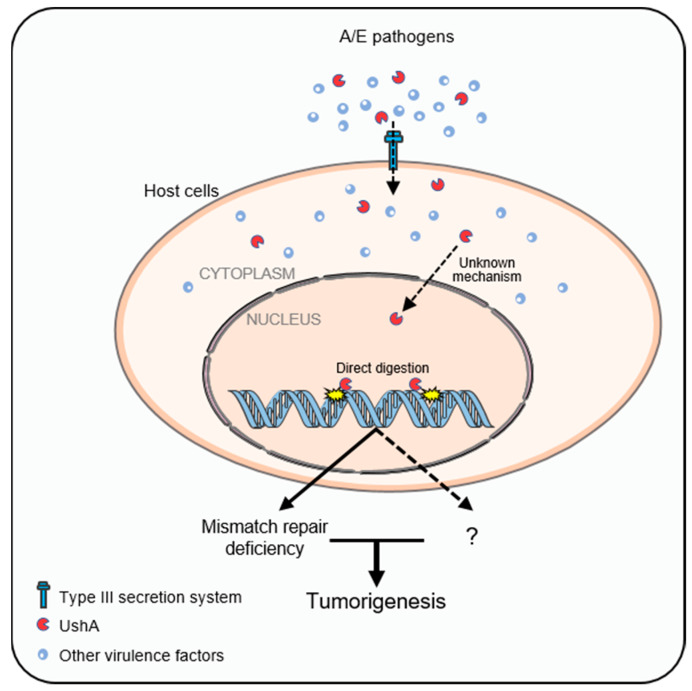
The novel genotoxin UshA and its role in tumorigenesis. A/E pathogens inject multiple virulence factors, including UshA, into host cells via the type III secretion system after colonization. The novel genotoxin UshA digests chromosomal DNA and results in unique mutational signatures in host cells, which is associated with DNA mismatch repair deficiency and tumorigenesis of CRC.

**Table 1 cancers-15-01152-t001:** The relationships between CDT-producing bacterial pathogens and cancers.

CDT-Producing Species	Related Cancer	Animal Model	Possible Contribution of CDT
*C. jejuni*	CRC [46]	Germ-free *Apc*^Min/+mice^	Promotes CRC and alters microbial composition and transcriptomic responses
*H. hepaticus*	Intestinal carcinoma [47]	129/SvEv Rag2−/− mice	Promotes intestinal carcinogenesis through inducing increased DSBs and activating the Stat3 signaling pathway
*H. hepaticus*	Hepatocarcinogenesis [48]	A/JCr mice	Promotes hepatocarcinogenesis by the activation of the NF-κB pathway

**Table 2 cancers-15-01152-t002:** The similarities and differences of *E. coli*-producing genotoxins.

Name of Genotoxin	Constitute	Function in DNA Damage	How to Get into Host Cells
**CDT**	CDT family are AB_2_ heterotrimers consisting of the active subunit CdtB (A) and two binding moieties, CdtA and CdtC (B) [25,26]	nicking or relaxation activities [30,39] DSBs [41,42] SSBs [43]	CDT binds to the receptor on the surface of host cell or is wrapped in outer membrane vesicles that can transport the toxin into the cell by fusion with the plasma membrane [28,29,30]
**Colibactin**	Colibactin is a polyketide-nonribosomal peptide [52,53]	stable DNA adducts [55,56] DSBs [52,57,58] replication stress [60] interstrand cross-links [60]	Unknown
**UshA**	UshA is a miltiple-functional enzyme [23,66,68]	DSBs +SSBs [23]	T3SS [23]

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
