# Peer review of "Genotoxins: The Mechanistic Links between Escherichia coli and Colorectal Cancer"

_cancers, 2023, doi:10.3390/cancers15041152_

Round 1

Reviewer 1 Report

This article is interesting and summarizes CDT, colibactin, and UshA. The structure of this article and the way it is discussed are very clear. Here, there are some suggestions:

1: Could the characteristics of these genotoxins be shown in a table format.

2: Can the authors discuss the similarities and differences of these genotoxins.

Reviewer 2 Report

Thank you for sending me the research article paper “Genotoxins: the mechanistic links between Escherichia coli and 2 colorectal cancer” for review in the Cancers. In the article of Wang et al., the author discussed the role of E. coli genotoxins in the development and progression of cancers. There are important points that should be discussed and improved.

1.      It would be better to divide the literature review into separate cancers heading. Explain regarding cancer types.

2.      Author should discuss the protein structure of cytolethal distending toxin in detail with a figure presentation.

3.      It would be good to discuss and predict the unknown mechanism of figure 2 and 3. Author should predict some possible mechanisms through figures.

4.      Author should present the strong linkage with cancer, as well as discuss the role of CDT in different cancers. (Tabular form presentation will be more understandable for readers.

Round 2

Reviewer 2 Report

Accepted